# An Angle Recognition Algorithm for Tracking Moving Targets Using WiFi Signals with Adaptive Spatiotemporal Clustering

**DOI:** 10.3390/s22010276

**Published:** 2021-12-30

**Authors:** Liping Tian, Liangqin Chen, Zhimeng Xu, Zhizhang Chen

**Affiliations:** 1School of Physics and Information Engineering, Fuzhou University, Fuzhou 350008, China; N181110016@fzu.edu.cn (L.T.); zhmxu@fzu.edu.cn (Z.X.); zdchen@fzu.edu.cn (Z.C.); 2Department of Electrical and Computer Engineering, Dalhousie University, Halifax, NC B3J 1Z1, Canada

**Keywords:** angle estimation, AOA, channel state information (CSI), DBscan, least-squares method, WiFi

## Abstract

An angle estimation algorithm for tracking indoor moving targets with WiFi is proposed. First, phase calibration and static path elimination are proposed and performed on the collected channel state information signals from different antennas. Then, the angle of arrival information is obtained with the joint estimation algorithm of the angle of arrival (AOA) and time of flight (TOF). To deal with the multipath effects, we adopt the DBscan spatiotemporal clustering algorithm with adaptive parameters. In addition, the time-continuous angle of arrival information is obtained by interpolating and supplementing points to extract the dynamic signal paths better. Finally, the least-squares method is used for linear fitting to obtain the final angle information of a moving target. Experiments are conducted with the tracking data set presented with Tsinghua’s Widar 2.0. The results show that the average angle estimation error with the proposed algorithm is smaller than Widar2.0. The average angle error is about 7.18° in the classroom environment, 3.62° in the corridor environment, and 12.16° in the office environment; they are smaller than the errors of the existing system.

## 1. Introduction

In recent years, indoor positioning technology has been developed and applied in many areas, and its commercial profits reached USD 10 billion in 2020 [1]. For example, it can help locate patients in a hospital and diagnose depression, mania, and so on. In home care and supervision of children, it can be used to monitor abnormal behaviors. In large warehouses, it can locate goods and valuables. It can also help rescue workers find trapped people in time in sudden disasters in industrial areas. As a result, various indoor positioning technologies have been developed. For example, the indoor positioning technology based on Bluetooth has been proposed [2,3], although its application is usually limited to a small range of about ten meters. The indoor positioning technology using ultrasonic waves has been presented in [4,5]. Nevertheless, indoor multipath has a great influence on positioning accuracy. An ultrasonic wave is susceptible to ambient temperature and the Doppler effect. The ultra-wideband technology has also been used for indoor positioning [6,7], but it is relatively expensive and has not been widely applied. The RFID-based indoor positioning technology has also been described [8,9,10], and it usually has a poor anti-jamming ability. Ref. [11] introduces an FPGA implementation of the position evaluation algorithm based on the TDOA principle. With the prevalence of WiFi signals, using them for indoor positioning has been studied and developed [12,13,14].

WiFi positioning and tracking algorithms can be divided into two types. The first is active positioning or tracking, such as SpotFi [15], Wicapture [16,17], and Milliback [18], but they are inconvenient because they require people to carry devices around with them.

The second technology is passive positioning or tracking. There are two main passive tracking algorithms: (1) fingerprint-based tracking algorithms [19,20,21,22] and (2) parameter-based indoor tracking algorithms. Fingerprint-based tracking algorithms collect a large number of samples in advance and use them for training an algorithm. They require a lot of energy and resources. In addition, they are highly dependent on the environments: the algorithms need to be recalibrated and retrained every time the environments change. On the other hand, the parameter-based approach does not require training and is independent of the environment but more computationally intensive.

The AOA is a key positioning parameter. Many algorithms have been developed for estimating AOA. SpotFi [15] collects CSI of WiFi signals and then applies the joint AOA estimation algorithm and the TOF to estimate the angle of a moving target. Dynamic Music applies the MUSIC algorithm of joint estimation of AOA and TOF in [23]. Widar 2.0 [24] uses a multiparameter estimation algorithm. By using a link in a group of RX-TX (three receiving antennas at each receiving end), the amplitude information, TOF information, AOA information, and Doppler velocity of the moving target signal can be estimated simultaneously. A median error of 0.75 m is achieved within the 8 m range. Because a four-dimensional search is required, the expectation-maximization algorithm is used to reduce the number of searches.

Static path refers to the signal reflected from stationary objects (such as furniture, walls, floors, etc.) to the receiver. A dynamic path is a signal that passes through a moving target and arrives at the receiving end. In general, we need to identify whether the AOA is a static or dynamic path [15]. Compared with other angle estimation algorithms, the static path elimination algorithm is first used in this paper, which eliminates the need to distinguish the static angle from the dynamic path angle in subsequent analysis. The above-mentioned methods use the instantaneous signals to estimate the AOA, not considering the history of the measured AOA of a moving target. To take advantage of the AOA history, this paper proposes an angle estimation algorithm that uses the past AOA information. In addition, phase recalibration and static path elimination are performed on the AOA-related CSI signal to remove the interferences of static or stationary objects. The DBscan spatiotemporal clustering algorithm is also adopted to mitigate the multipath problem. Finally, the least-squares method is used for linear fitting to obtain the final angle information of the moving target.

In short, the main contributions of this paper are:(1)Phase calibration and static path elimination are performed on the collected CSI signals, and then AOA and TOF are jointly used for the AOA estimations;(2)A DBscan spatiotemporal clustering algorithm with adaptive parameter adjustment is proposed to reduce multipath effects;(3)The linear fitting method of the least-squares method is introduced and applied to supplement and finalize the AOA results.

Note that in our studies, we use the tracking data set of Tsinghua’s Widar2.0 to test our proposed algorithm. This data set includes 24 trajectories in classrooms, offices, and corridors and has about 1700 pieces of angle information.

This article will discuss it in detail in the Materials and Methods (Section 2), Results (Section 3), and Conclusions (Section 4).

## 2. Materials and Methods

In the following subsections, the proposed algorithm is elaborated. It includes CSI model building, phase calibration, static path elimination, and the angle estimation algorithm jointly with AOA and TOF, the spatiotemporal clustering algorithm, and the least-squares linear fitting shown in Figure 1.

### 2.1. CSI Model

As described before, the WiFi signals propagate in space and are scattered by any objects they encounter in an indoor environment. Therefore, the WiFi signals’ CSI embodies the information about static and dynamic objects (and thus paths) in an indoor environment.

Consider the receiving array as shown in Figure 2. It has *M* elements.

Assume the *i*th subcarrier signal received by the *m*th antenna element is h(i,m,t). It can be expressed as:(1)$$h(i,m,t)=\sum_{l=1}^{L}a_{l}(i,m,t)e^{-j2\pi f_{i}\tau_{l}(i,m,t)}+N(t)$$
where *L* represents the total number of paths. *N*(*t*) represents noise in the path. $a_{l}(i,m,t)$ represents the amplitude of the *i*th subcarrier signal received by the *m*th element along path *l*. $\tau_{l}$ represents the signal flight time along path *l*. If the phase of the 1th subcarrier signal h(1,1,t) of 1th antenna is taken as the reference phase, the phase difference of the *i*th subcarrier h(i,m,t) of the *m*th antenna with respect to h(1,1,t) can be expressed as:(2)$$2\pi f_{i}\tau_{l}(i,m,t)=2\pi (\Delta f_{i}\tau_{l}+f_{i}d\cdot (m-1)\cdot \frac{\sin(\phi_{l})}{c})$$
where $\Delta f_{i}$ is the frequency difference between the subcarrier *i* and the reference carrier, *d*(*m* − 1) is the extra propagation distance between the *m*th antenna and the reference antenna, *c* is the speed of light, and $\phi_{l}$ is the AOA of path *l*.

As seen, phase differences (Equation (2)) between adjacent antennas contain AOA information. However, due to imperfect hardware clock synchronization, time offset, frequency shifts, and initial phases can cause measurement errors. Therefore, phase calibration is required as described below. In addition, we are interested in tracking moving targets and then the dynamic phase. We then need to remove the static path information, specifically in AOA determination. They are elaborated in the following subsections.

### 2.2. Phase Calibration and Static Path Elimination

Since there is no strict clock synchronization in the CSI signal reception, there will be an error between the measured CSI signal $\tilde{h}(i,m,t)$ and the actual CSI signal h(i,m,t):(3)$$\tilde{h}(i,m,t)=h(i,m,t)e^{-2\pi j(\Delta f_{i}\varepsilon_{t}+\Delta t\cdot \varepsilon_{{}_{f}})+\zeta_{s}}$$
where $\varepsilon_{t}$ is the time offset, $\varepsilon_{f}$ is the frequency offset, and $\zeta_{s}$ is the initial phase offset.

Since the time offset and frequency offset are the same for the signals received at different antenna elements, they can be removed by conjugate multiplication [24,25]. Conjugate multiplication of signals of one set of antennas and other antenna signals can be obtained:(4)$$\begin{array}{l} T(i,m,t)=angle(\tilde{h}(i,m,t))\times {\tilde{h}}^{*}(i,m_{0},t)\\ =angle(h(i,m,t))\times h^{*}(i,m_{0},t) \end{array}$$
where $m_{0}$ is the reference antenna and m is for all antennas. $angle(\tilde{h}(i,m,t))$ means the phase of $\tilde{h}(i,m,t)$, shown as follows:(5)$$angle(\tilde{h}(i,m,t))=\exp(j\cdot \varphi (\tilde{h}(i,m,t)))$$
h(i,m,t) and $h(i,m_{0},t)$ are expressed as follows:(6)$$h(i,m,t)=\sum_{ls=1}^{Ls}h_{ls}(i,m,t)+\sum_{ld=1}^{Ld}h_{ld}(i,m,t)$$
and
(7)$$h(i,m_{0},t)=\sum_{ls=1}^{Ls}h_{ls}(i,m_{0},t)+\sum_{ld=1}^{Ld}h_{ld}(i,m_{0},t).$$

Substitution of Equations (6) and (7) into Equation (4) reads:(8)$$\begin{array}{l} T(i,m,t)=\sum_{ls=1}^{Ls}\left(\exp(j\cdot \varphi (h_{ls}(i,m,t\right)))\cdot h_{ls}{}^{*}(i,m_{0},t))+\\ \sum_{ls=1}^{Ls}\left(\exp(j\cdot \varphi (h_{ls}(i,m,t\right))))\cdot \sum_{ld=1}^{Ld}(h_{ld}{}^{*}(i,m_{0},t))+\\ \sum_{ld=1}^{Ld}\left(\exp(j\cdot \varphi (h_{ld}(i,m,t\right))))\cdot \sum_{ls=1}^{Ls}(h_{ls}{}^{*}(i,m_{0},t))\\ +\sum_{ld=1}^{Ld}\left(\exp(j\cdot \varphi (h_{ld}(i,m,t)))\cdot h_{ld}{}^{*}(i,m_{0},t\right)) \end{array}$$

The $\sum_{ls=1}^{Ls}\exp(j\cdot \varphi (h_{ls}(i,m,t)))\cdot (h_{ls}{}^{*}(i,m_{0},t))$ is caused by the static path and is of low frequency. $\sum_{ld=1}^{Ld}(\exp(j\cdot \varphi (h_{ld}(i,m,t)))\cdot h_{ld}{}^{*}(i,m_{0},t))$ is caused by the dynamic path and is of high frequency. The above two terms can be removed with a bandpass filter. The remaining two terms are expanded as follows:(9)$$\begin{array}{l} \exp(j\cdot \varphi (h_{ls}(i,m,t))\cdot h_{ld}{}^{*}(i,m_{0},t)=a_{ld}{}^{*}\exp(-j2\pi \\ \left(\Delta f_{i}(\tau_{ls}-\tau_{ld})+f_{c}d(m-m_{0})(\sin(\phi_{ls})-\sin(\phi_{ld}))/c\right) \end{array},$$
and
(10)$$\begin{array}{l} \exp(j\cdot \varphi (h_{ld}(i,m,t))\cdot h_{ls}{}^{*}(i,m_{0},t)=a_{ls}{}^{*}\exp(-j2\pi (\Delta f_{i}\\ \left(\tau_{ld}-\tau_{ls})+f_{c}d(m-m_{0})(\sin(\phi_{ld})-\sin(\phi_{ls}))/c\right) \end{array}$$

By comparing Equations (10) and (11), we can get:(11)$$\begin{array}{l} \left|\exp(j\cdot \varphi (h_{ld}(i,m,t))\cdot h_{ls}{}^{*}(i,m_{0},t)\right|=\left|a_{ls}\right|\gg \\ \left|a_{ld}\right|=\left|\exp(j\cdot \varphi (h_{ls}(i,m,t))\cdot h_{ld}{}^{*}(i,m_{0},t)\right| \end{array}$$

Since the amplitude of the static path signal $\left|a_{ls}\right|$ is far greater than that of the dynamic path signal $\left|a_{ld}\right|$ (In the case that there is no occlusion between the transmitting and receiving antennas), the latter can be neglected. We did an experiment where a person moves away from an antenna and then slows down to a stop and vice versa. The results of the short-time Fourier transform (STFT) before and after static elimination are shown in Figure 3a,b. In the absence of static cancellation, noise, dynamic path signal, and static path signal are mixed together and cannot be distinguished. After the static elimination, there is little noise, and the signal contains mostly the dynamic path signal. This is very helpful for extracting accurate AOA information later.

To obtain a more accurate measurement of phase information and the dynamic path, we further apply the well-known minimum-norm method (MNM) [26] signal processing algorithm to estimate AOA.

### 2.3. Joint Estimation with AOA and TOF Using the MNM Algorithm

The mathematical expression of a narrowband far-field signal is:(12)h(i,m,t)=A⋅s(i,m,t)+N(t)
where h(i,m,t) is the received signal, A is the guiding vector of the antenna array, s(i,m,t) is the signal matrix, N(t) is the noise signal, and the unbiased estimate of the covariance matrix of the signal is
(13)$$R(i,m,t)=\frac{1}{N}\sum_{n=1}^{N}h_{n}(i,m,t)h_{n}{}^{H}(i,m,t)$$
where *n* is the number of snapshots, and *H* is the transrank operator.

Substitution of (12) into (13) reads:(14)$$R=E(h\cdot h^{H})=AE(S\cdot S^{H})A^{H}+E(N\cdot N^{H})$$
where *E* is the expectation operator.

However, three antennas can only identify two paths at most. For a typical indoor signal of 6–8, three antennas are not enough. However, each antenna receives 30 subcarriers, so the phase deviation caused by different subcarriers in different paths of TOF is used to extend the antenna [14].

The phase shift caused by the introduction of path *ld* in adjacent different antennas is $d\sin(\theta_{ld})/c$. The phase shift in the *M*th antenna of the propagation path *ld* relative to the reference antenna 1 is
(15)$$\Phi (\theta )=e^{-j2\pi d(M-1)\sin(\theta_{ld})\times f/c}$$

The steering vector of the signal in path *ld* caused by the antenna is:(16)$$\vec{a}(\theta_{ld})={\left[1\;\Phi (\theta_{ld})\ldots \Phi {\left(\theta_{ld}\right)}^{\left(M-1\right)}\right]}^{T}$$

The steering vector for all multipath paths is
(17)$$\vec{a}(\theta )={\left[\vec{a}(\theta_{1}),\vec{a}(\theta_{2}),\cdots ,\vec{a}(\theta_{Ld})\right]}^{T}$$

The phase shift caused by the introduction of the *ld* path in different adjacent subcarriers is $2\pi (f_{i}-f_{i+1})\tau_{ld}$.

The phase shift introduced by the time of flight of path *ld* to the first subcarrier of the same antenna at the *i*th subcarrier is $2\pi (i-1)f_{\delta }\tau_{ld}$, where $f_{\delta }$ is the frequency interval between two continuous subcarriers. Denote
(18)$$\phi (\tau_{ld})=e^{-2\pi (i-1)f_{\delta }\tau_{ld}}$$

The signal guidance vector of path *ld* with the signal composed of the subcarrier and the antenna is:(19)$$\begin{array}{l} \vec{a}(\theta_{ld})=[\overset{\mathrm{antenna}1}{\overbrace{1\;,\Phi (\tau_{ld}),\cdots ,\Phi {\left(\tau_{ld}\right)}^{I-1}}},\\ \underset{\mathrm{antenna}2}{\underbrace{\Phi (\theta_{ld}),\cdots ,\Phi (\theta_{ld})\Phi {\left(\tau_{ld}\right)}^{I-1}}},\cdots ,\\ \overset{\mathrm{antenna}M}{\overbrace{\Phi {\left(\theta_{ld}\right)}^{\left(M-1\right)},\cdots ,\Phi {\left(\tau_{ld}\right)}^{I-1}\Phi {\left(\theta_{ld}\right)}^{\left(M-1\right)}}}{]}^{T} \end{array}$$

The guiding vector for all multipath paths is
(20)$$\vec{A}={\left[\vec{a}(\theta_{1}),\vec{a}(\theta_{2}),\cdots ,\vec{a}(\theta_{Ld})\right]}^{T}.\;$$

The MNM algorithm is a subspace algorithm with constrained weights. The MNM algorithm has a higher resolution than the MUSIC algorithm, as shown in Figure 4a,b.

The MNM algorithm constraint conditions are as follows:(21)$$\{\begin{array}{c} \min\left\{W^{H}W\right\}\\ W(1)=1,U_{s}W=0 \end{array}$$
$U_{s}$ is the signal space obtained by Formula (15). *W* is a linear combination of the noise space. Estimates of TOF are obtained by looking for the peak of the formula:(22)$$p_{MNM}=\frac{1}{{\left|A^{H}W_{MNM}\right|}^{2}}$$

Figure 5 shows the result of AOA obtained by joint estimation of AOA and TOF after phase calibration and path elimination. It can be seen that multipath signals are very complex.

### 2.4. Spatiotemporal Clustering Algorithm of DBscan Based on Adaptive Parameter Adjustment

Figure 5 shows that it is difficult to find the angle information of the moving target from the MNM algorithm. Meanwhile, DBscan is a density clustering algorithm. It does not need to specify the center of the cluster and the number of clusters. Since the angle information changes continuously in the adjacent time, we cluster the angle information together with the time information.

To ensure the consistency of spatiotemporal information, we add the time-varying scale parameter ρ. New time Unit change is:(23)$$t^{\prime }=\rho \cdot t.$$

With *q* as the center, the number of points in the *p* neighborhood whose radius is smaller than *Eps* is
(24)$$N_{Eps}(p)=\left\{q\in D|dist(p,q)\leq Eps\right\}$$
where *Eps* is the radius of the cluster circle, *dist* is the distance formula, and it is defined as follows:(25)$$dist(p,q)=\sqrt{{\left(AOA_{p}-AOA_{q}\right)}^{2}+\rho^{2}{\left({t_{p}}^{\prime }-{t_{q}}^{\prime }\right)}^{2}}$$

If the number of elements in the *p* neighborhood is less than the specified parameter *Minpts*, it is considered a noise point. If the number of points is greater than or equal to it, a new cluster is created, and the point is added to that cluster. *Eps* parameters directly affect clustering results and generally need to be adjusted experimentally. Here, we use adaptive parameter adjustment and the specific algorithm as Algorithm 1.
**Algorithm 1**: Spatiotemporal Cluster Algorithm **Input**: *D*, *Eps0*, *t*′, *Minpts*, *t*_0_**Output**: *clu* While (*clu*(*t′*(end))-*clu*(*t′*(start) > (*size*(*t*)-*t*_0_))  Do **step 1**: The distance distribution of the points to be clustered is calculated     **step 2**: for *i* = 1:*n**     Neighbors* = find(*dist*(*D(i)*) ≤ *Eps0*)      If *num (Neighbors*) < *Minpts*       *D(i)* = noise       Else Expand Cluster (*D(i)*, *Neighbors*)      End if      End forEnd while

Figure 6 shows the results of DBscan clustering after adaptive parameter adjustment. The black point is the noise information, and the clustering result of the red point with the most points is the AOA result of the dynamic target we need. It can be seen from the figure that clustering results can remove a lot of noise and multipath information.

### 2.5. Processing of AOA Data after Clustering

The AOA after clustering is not continuous in time. Because the initial velocity of the target is relatively small when the target starts to move, the dynamic path obtained by the static path elimination algorithm may not be detected. In addition, dynamic paths are unstable. Therefore, the undetected points need to be supplemented to ensure their continuity in time. For those with multiple clustering angles at the same time, take their mean values; for those with missing values at the starting time point, fill in at the adjacent points. If the intermediate time point is missing, the mean of the time points before and after is added.

After the above steps, AOA is continuous in time, but the fluctuation is relatively large, which is not in line with reality. So, we used the polynomial least-squares linear fitting. At *t*_1_, *t*_2_, …, *t_Z_* time, the AOA values are *AOA*_1_, *AOA*_2_, …, *AOA_Z_*. Let us write it as a function *AOAz* = *f*(*t_Z_*). The target polynomial of the fit is
(26)$$f(t_{Z})=a_{0}+a_{1}t_{Z}+a_{2}t_{Z}{}^{2}+\cdots +a_{n}t_{Z}{}^{n}.$$

The fitting process is an optimization problem. Let the sum of mean square error between the fitted polynomial value and the original function value be minimized. The mathematical expression is as follows:(27)$$\sum_{z=1}^{Z}{\left(f(t_{z})-AOA_{z}\right)}^{2}=min.$$

Figure 7 shows the results of polynomial fitting using the least-squares method. The red asterisk is the result of *AOA* after interpolation. The blue line is the polynomial linear fitting results of the least-squares method.

## 3. Results

This section mainly includes two parts: experimental setting, environment introduction, and experimental result analysis.

### 3.1. Experimental Setting and Environment

We used the Widar2.0 data set. A sending antenna and three receiving antennas are used in the experiment. The receiving antennas are placed half a wavelength apart. The antenna used in this paper is a uniform linear array. The antenna is for the WiFi protocol 5300 network card 2.4 G and 5 G dual-band antenna. The device is operating in monitor mode. The number of the channel is 165, which has a center frequency of 5.825 GHz. The transmitting antenna has a packet rate of 1000 packets per second. There are three experimental environments: a large and empty classroom, a small office with lots of furniture, and a long and narrow corridor.

There are 24 trajectories in the three environments of Widar 2.0. There are about 70 angles for each trajectory and about 1700 angle information in total. In the 24 trajectories, there are 6 that take the shape of letter ‘Z’ (facing 3 different directions), 7 circles (starting points at 4 different positions), 2 symmetrical paths of number ‘7’, 1 rectangle, 6 vertical lines (2 different starting points), and 2 oblique lines (2 different starting points).

### 3.2. Analysis of Experimental Results

Firstly, the classroom environment data are analyzed. These data use T02 data of the classroom environment. The trajectory is in the direction perpendicular to the cable line of the transmitting and receiving antennas, first approaching and then backward away. A position is estimated every 0.1 s for 4.6 s, and 46 angles are estimated.

The angle estimation results are shown in Figure 8, where the red line is the actual angle information. The blue line is the angle estimation result with the proposed algorithm. The yellow line is the angle estimation results obtained with Tsinghua’s Widar2.0. The average angle error of the proposed algorithm is 5.14°, and the average angle error of Widar2.0 is 8.53°. Because the algorithm in this paper is based on temporal and spatial clustering and fully considers the continuity of angle information of the adjacent time, the average error with the proposed algorithm is relatively small, and the accuracy is relatively high.

Here are the results of the T02 data for the office environment. The trajectory is perpendicular to the cable line of the transmitting and receiving antennas, first approaching and then backward away. Position estimation is made every 0.1 s. The time of the trajectories is 5.6 s, with 56 angles estimated. The angle estimation results are shown in Figure 9: the red line is the actual angle information, and the blue line is the angle estimation result obtained with the proposed algorithm. The yellow line is the angle estimation results of Tsinghua’s Widar2.0. The average angle error of the algorithm in this paper is 12.14°, and the average angle error of Widar2.0 is 16.26°. Because the algorithm in this paper is based on temporal and spatial clustering and fully considers the continuity of angle information of the adjacent time, the average error of the proposed algorithm is relatively small, and the accuracy is relatively high.

The following are the T01 data obtained in the corridor environment. The trajectory is perpendicular to the cable line of the transmitting and receiving antennas, first, moving away and then coming back. Position estimation is made every 0.1 s. The time of the trajectories is 6.7 s, with 67 angles estimated. There are 67 angles. The angle estimation results are shown in Figure 10, where the red line is the actual angle information. The blue line is the angle estimation result of the proposed algorithm. The yellow line is the angle estimation results of Tsinghua’s Widar2.0. The average angle error of the proposed algorithm is 1.7°, and the average angle error of Widar2.0 is 4.5°. Because the algorithm in this paper is based on temporal and spatial clustering and fully considers the continuity of angle information of the adjacent time, the average error of the proposed algorithm is relatively small, and the accuracy is relatively high.

The data with only one trajectory may not be so complete. We ran all data for the three environments of the classroom, office, and corridor, respectively. The cumulative angle error of the corridor is shown in Figure 11. In Figure 11, the blue line is the error accumulation of the proposed algorithm, and the red line is the error accumulation curve of Tsinghua Widar2.0. It can be seen from the figure that 90% of the angle error of the proposed algorithm in this paper is less than 7.74°, and 90% of the angle error of the Tsinghua Widar2.0 algorithm is less than 10.72°. The algorithm in this paper is slightly better than the latter.

The cumulative angle error of the office is shown in Figure 12. In Figure 12, the blue line is the error accumulation curve of the proposed algorithm, and the red is the error accumulation curve of Widar2.0. It can be seen from the figure that 90% of the angle error of the algorithm is less than 20.63°, and 90% of the angle error of the Tsinghua Widar2.0 algorithm is less than 25.61°. The algorithm in this paper is slightly better than the latter.

We ran the data for all the tracks in the classroom environment. The angle error accumulation diagram is shown in Figure 13. In Figure 13, the blue line is the error accumulation curve of the algorithm in this paper, and the red line is the error accumulation curve of Widar2.0. It can be seen from the figure that 90% of the angle error of the algorithm in this paper is less than 13.68°, and 90% of the angle error of the Tsinghua Widar2.0 algorithm is less than 15.83°. The algorithm in this paper is slightly better than the latter.

### 3.3. System Performance

What factors will affect the accuracy of the proposed algorithm? In order to understand the system performance of this algorithm, we analyze from different environments, different walking speeds, different data sampling rates, different trajectories shapes, different walking directions, different lengths of the trajectories, and different filtering methods.


(1)Different environments.


As known, indoors is a typical multipath environment. Multipath information will affect the accuracy of our parameter estimation. In this experiment, we analyzed the errors of the three test environments, respectively. Figure 14 is the error accumulation function for the three environments. From the figure, we can find that the average error of AOA in the corridor environment is the smallest. Because there is no furniture in the corridor, the multipath interference is less, and the result is relatively more accurate. In the office environment, there are sofas, tea tables, drinking fountains, and storage cabinets. They will cause more severe multipath and relatively larger errors. In the classroom environment, only desks are placed beside the wall. The degree of multipath effects lies in between that with the corridor and that with the office, and so do the errors.


(2)Different walking speeds.


In order to understand the impact of walking speed on the system performance, we conduct the following experiments. The errors of AOA are analyzed under three different conditions of the target: relatively slow walking speed, normal walking speed, and fast walking speed. The results are shown in Figure 15. As we can see, the error is relatively large when the target walks slowly. This is because we apply the static path elimination beforehand. When the target walks slowly, part of useful information will be eliminated, so the error is large. On the other hand, when the target walks faster, less useful information is eliminated, and the error is smaller.


(3)Different sampling rates.


In our case, 1000 data packets are collected every second. If the sampling rate is small, what will happen? For this reason, we compare the error in the case of the sampling rates of 500/s and 1000/s, as shown in Figure 16. As the sampling rate decreases, the error increases, as we would expect.


(4)Different shapes’ trajectories.


Will different trajectories affect the estimation of AOA? For this reason, we compared the errors under three different trajectory shapes: z-shaped route, rectangular close loop, and vertical line. As can be seen from Figure 17, the overall error of the vertical line is the smallest. This is because when you go straight, the angle information is more continuous, and the result is more accurate. For the rectangular loop, there is a significant angle change at the turning corners, so the error is the largest. For the z-shape route, the error lies in between that with the rectangular and the vertical.


(5)Different directions of motion.


In order to understand the impact of different motion directions on the algorithm in this paper, we analyze the two cases of the walking direction and receiving and receiving line being 90° and 45°.

The results are shown in Figure 18. Because the AOA of the receiving antenna varies less along the direction of 45°, the corresponding results are more accurate.


(6)Different walking distances.


Does the length of the walk distances affect the angle estimation? We conduct the following experiment to analyze the errors under different walking distances. As shown in Figure 19, distance has no impact on angle estimation. Since the speed is relatively low at the beginning and end of the walk, the error between distances of 0–3 m and 9–13 m is large, which is consistent with the previous analysis.


(7)Different filtering methods.


In this paper, two methods are used to compare the final processing of AOA: one is the least-squares linear fitting, the other is the multiple uses of Hampel and smooth filtering. The result is shown in Figure 20. The linear fitting can fit the original results to the maximum extent. The smoothing and Hampel can smooth out the places where the angle changes dramatically, resulting in the loss of details. Therefore, the linear fitting method is more accurate.

## 4. Conclusions

This paper presents a new angle recognition algorithm for moving targets. In this algorithm, phase noise and static path are eliminated by conjugate multiplication of CSI signals of different antennas. Then, AOA and TOF are used to estimate the AOA value of the moving target. Next, the spatial and temporal clustering algorithm adjusted by adaptive parameters is used to obtain a more accurate AOA. Then, the continuous AOA value is obtained by the method of difference complementation. Finally, the least-squares polynomial linear fitting method is used to obtain the final AOA value. Through the verification of all the tracks in three different environments of Widar2.0, the average errors of the proposed algorithm in the classroom, office, and corridor are found to be 7.18°, 12.16°, and 3.62°, respectively. The average errors of Widar2.0 are 7.49°, 16.42°, and 4.55 °, respectively. Therefore, the proposed algorithm is a better algorithm. There are three experimental environments for the algorithm in this paper: classroom, office, and corridor. This paper does not consider how to model in the case of obstacles and occlusion. For both cases, references are available [27]. The literature points out that the uniqueness of each propagation path, where paths can be clustered on the basis of obstacles, whose dimensions are larger than the wavelength involved, where these clusters can be described by an appropriate shadow depth (namely, the shadow deviation), providing a log-additive expression of shadow losses (excess path loss, in general) in the RF formulas dealing with local mean power estimation/prediction. These would also be complementary to localization issues in indoor environments, as shown in [28]. The respective presence of shadow (large-scale fading) largely influences the CDFs of the received signal for each and every antenna, as shown in [29]. This will be our follow-up research work.

## Figures and Tables

**Figure 1 sensors-22-00276-f001:**
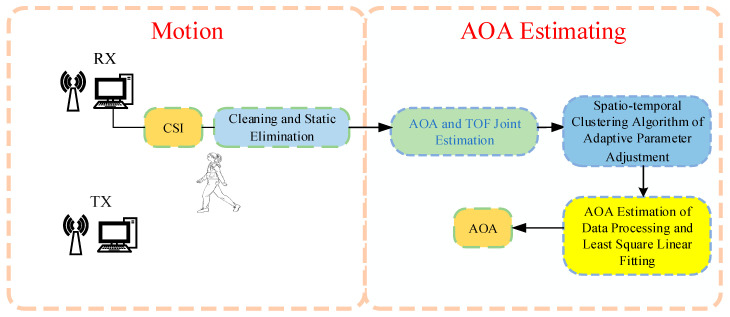
Algorithm flow chart (RX is the receiving antenna and TX is the transmitting antenna).

**Figure 2 sensors-22-00276-f002:**
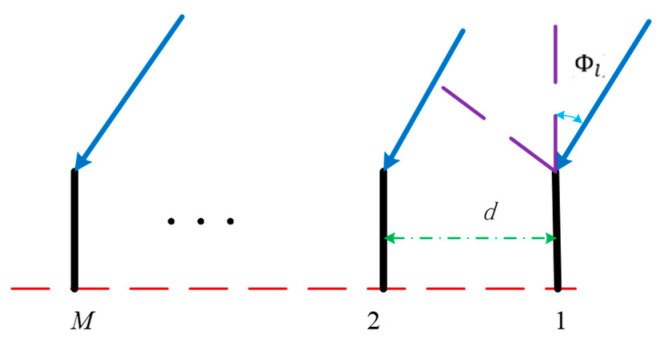
Receiving array (*d* is the array spacing, $\phi_{l}$ is the AOA of path *l* and *M* is the *M*th array antenna).

**Figure 3 sensors-22-00276-f003:**
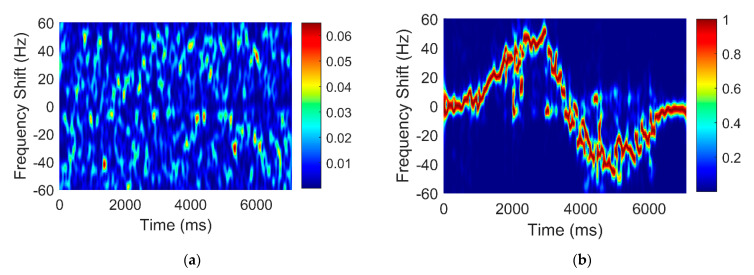
(**a**) Measured time-frequency chart without static elimination; (**b**) Measured time-frequency chart with static elimination.

**Figure 4 sensors-22-00276-f004:**
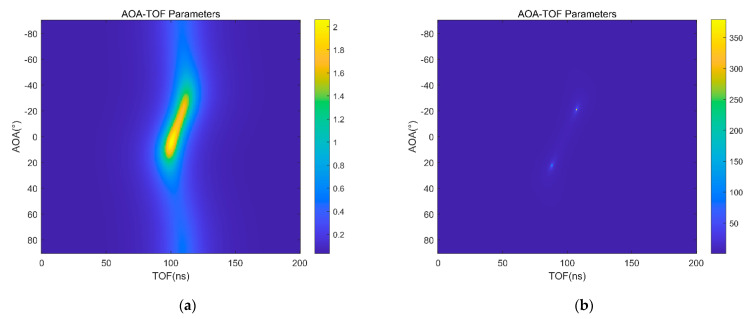
(**a**) AOA and TOF are estimated by MUSIC algorithm; (**b**) AOA and TOF are estimated by MNM algorithm.

**Figure 5 sensors-22-00276-f005:**
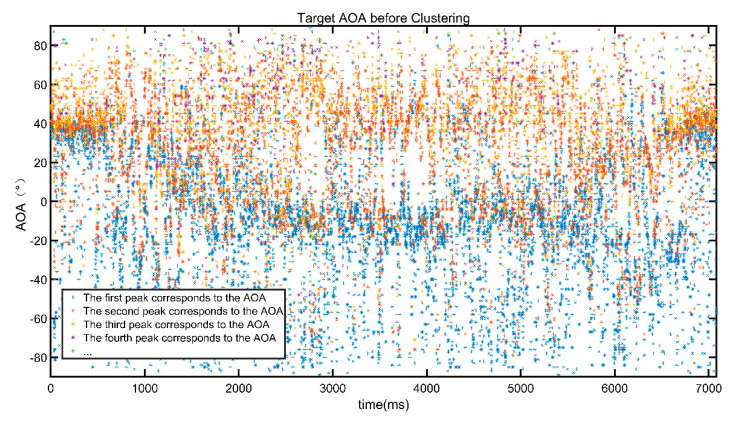
AOA value of the dynamic target after joint estimation.

**Figure 6 sensors-22-00276-f006:**
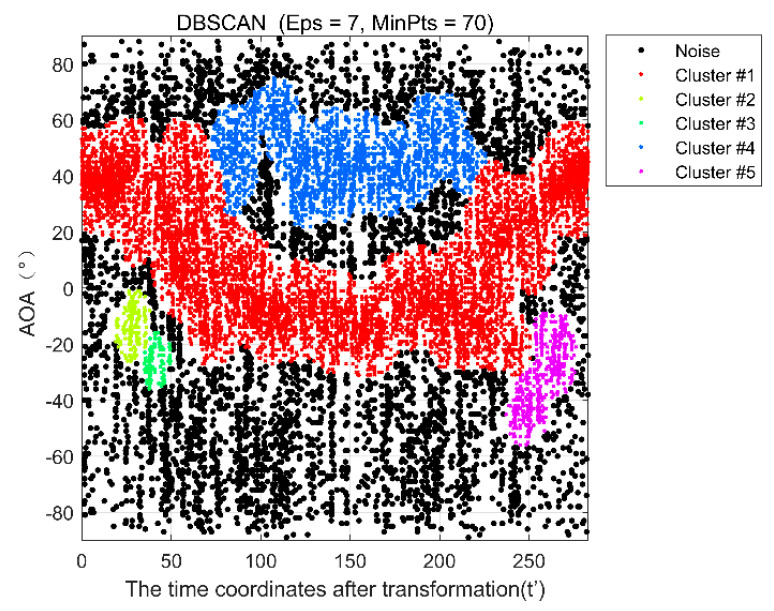
AOA spatiotemporal clustering results with adaptive parameters.

**Figure 7 sensors-22-00276-f007:**
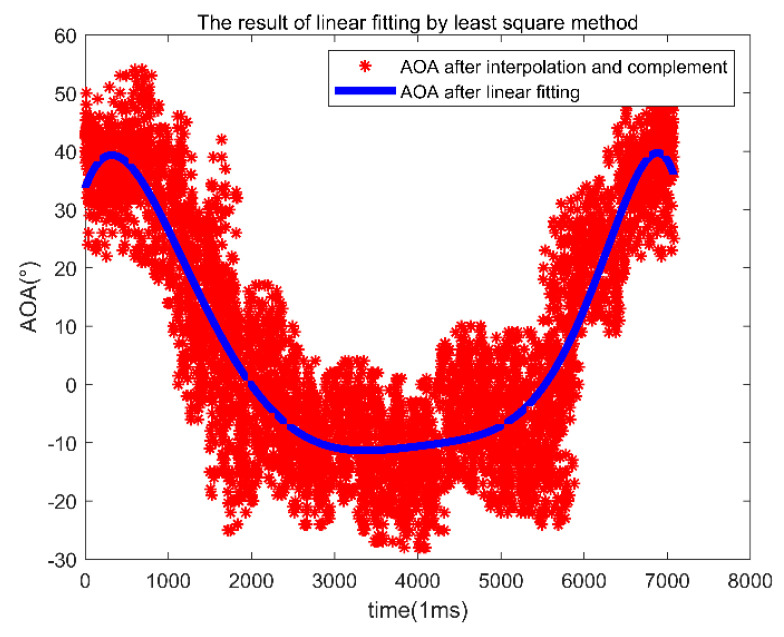
AOA results of linear fitting by least-squares method after value filling points.

**Figure 8 sensors-22-00276-f008:**
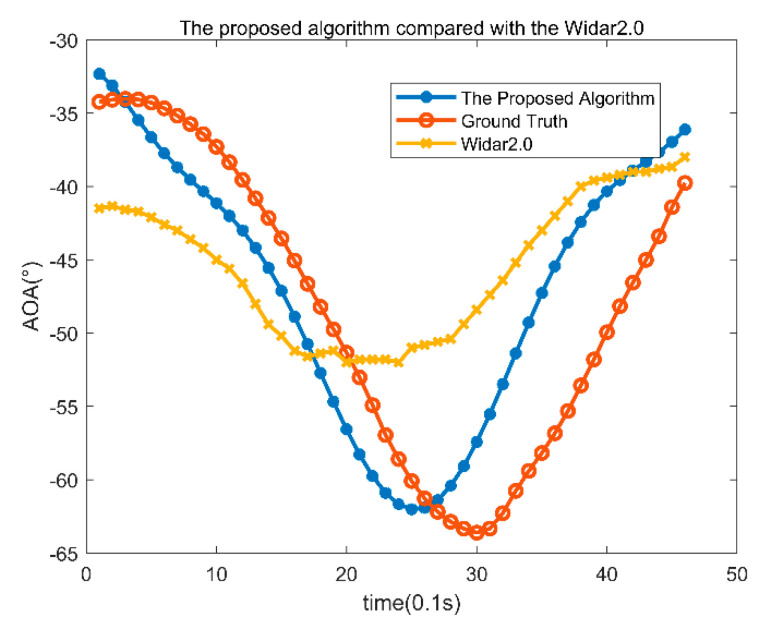
AOA results of classroom T02 data.

**Figure 9 sensors-22-00276-f009:**
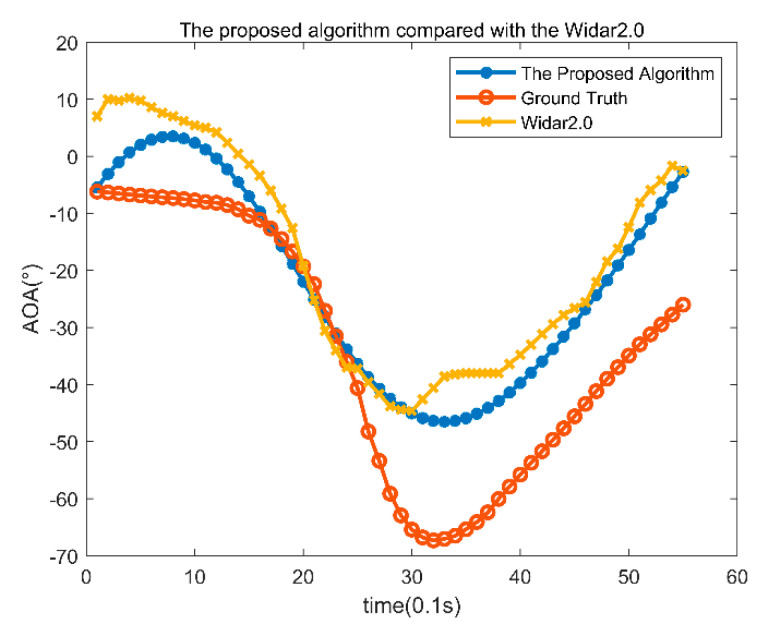
AOA results of office T02 data.

**Figure 10 sensors-22-00276-f010:**
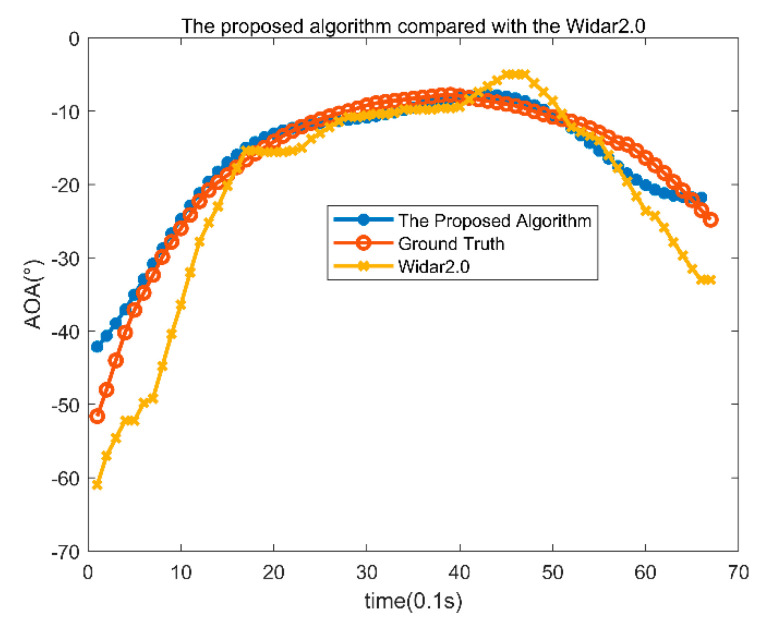
AOA results of corridor T01 data.

**Figure 11 sensors-22-00276-f011:**
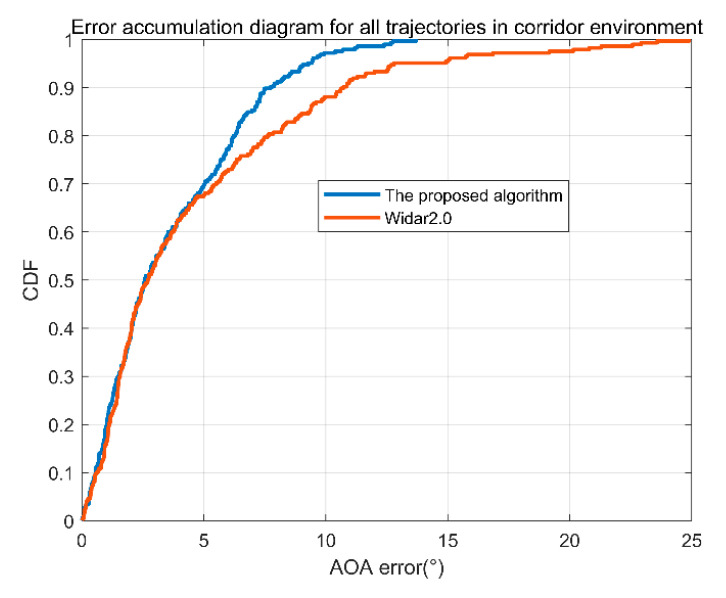
Cumulative angle error of the corridor.

**Figure 12 sensors-22-00276-f012:**
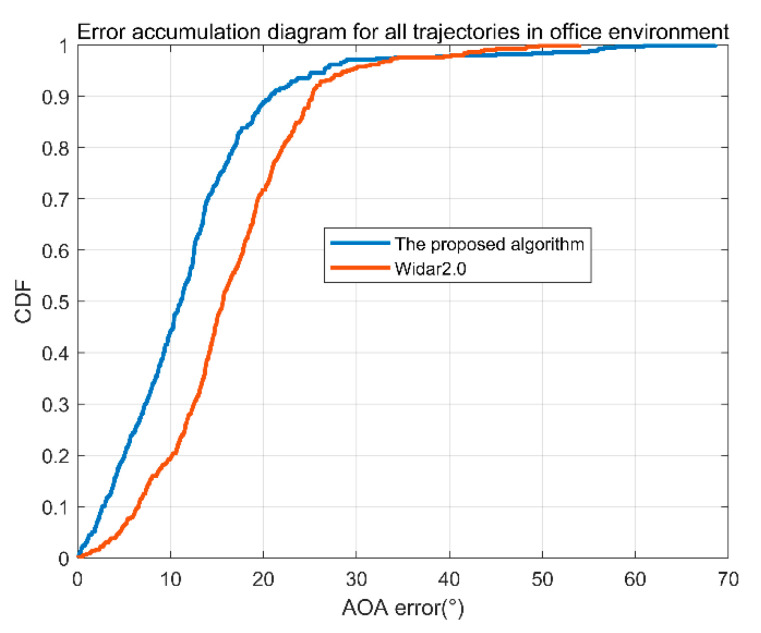
Cumulative angle error of the office.

**Figure 13 sensors-22-00276-f013:**
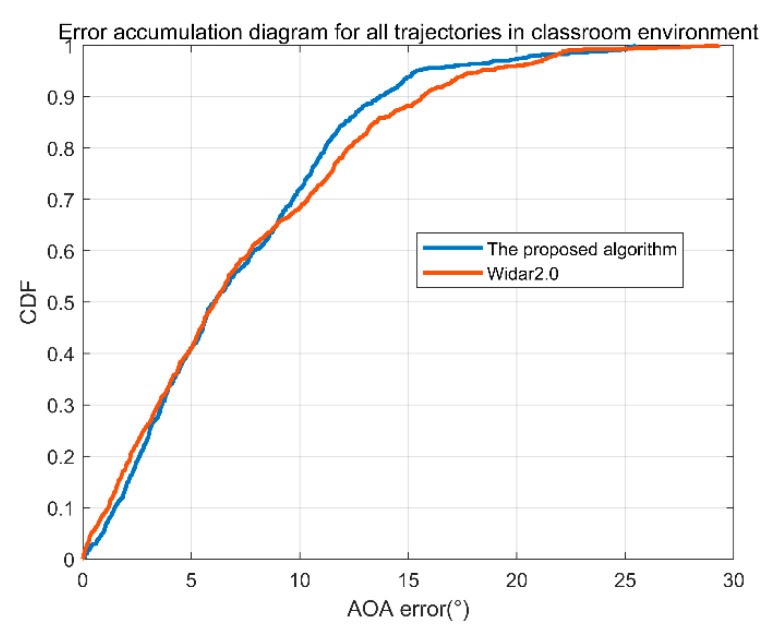
Cumulative angle error of the classroom.

**Figure 14 sensors-22-00276-f014:**
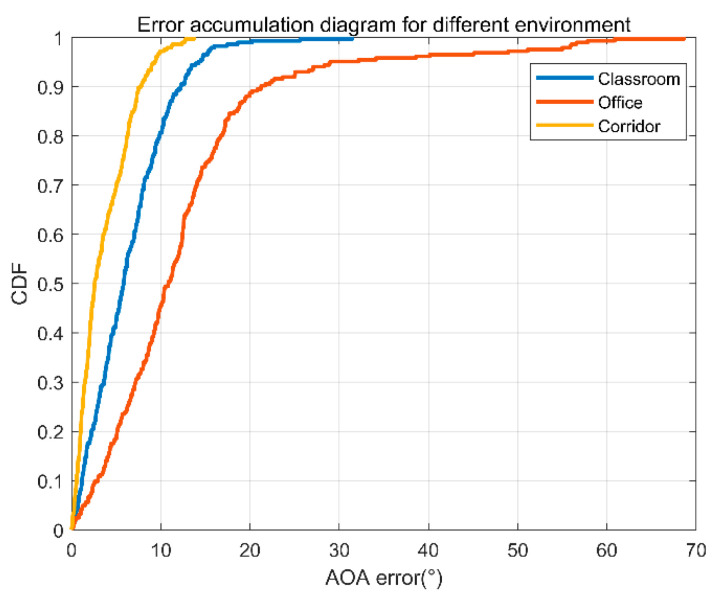
AOA error accumulation diagram of three different environments.

**Figure 15 sensors-22-00276-f015:**
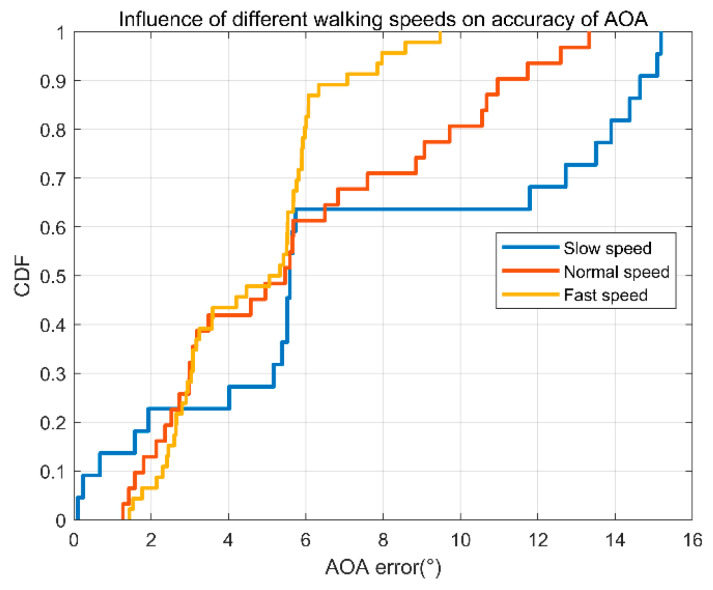
AOA error accumulation diagram of three different walking Speeds.

**Figure 16 sensors-22-00276-f016:**
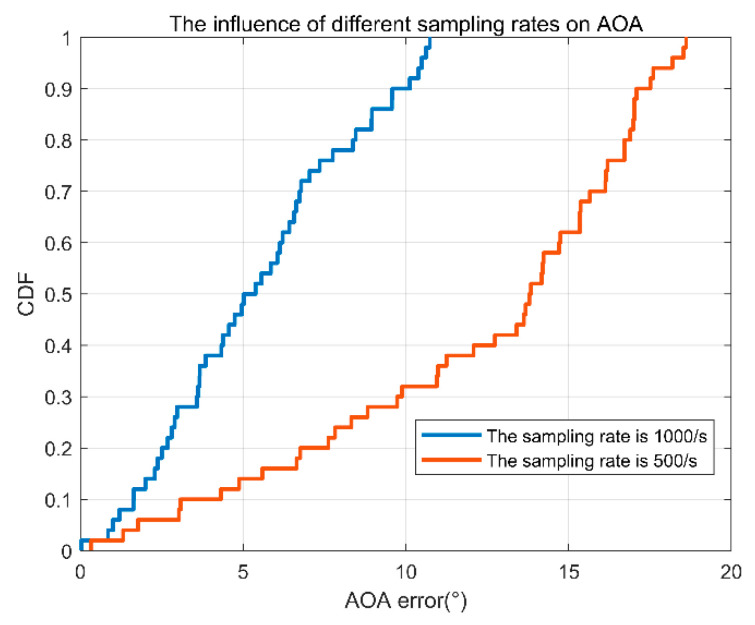
AOA error accumulation diagram of two different sampling rates.

**Figure 17 sensors-22-00276-f017:**
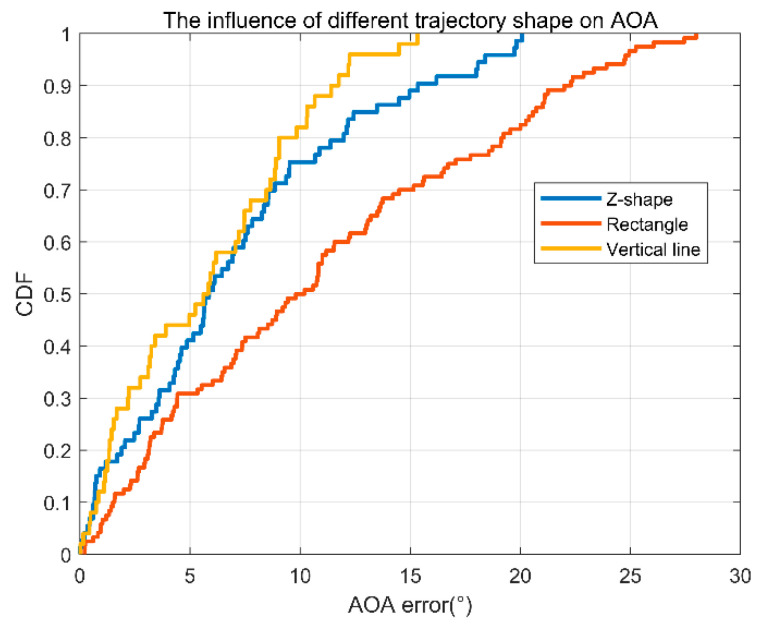
AOA error accumulation diagram of three different shapes’ Trajectories.

**Figure 18 sensors-22-00276-f018:**
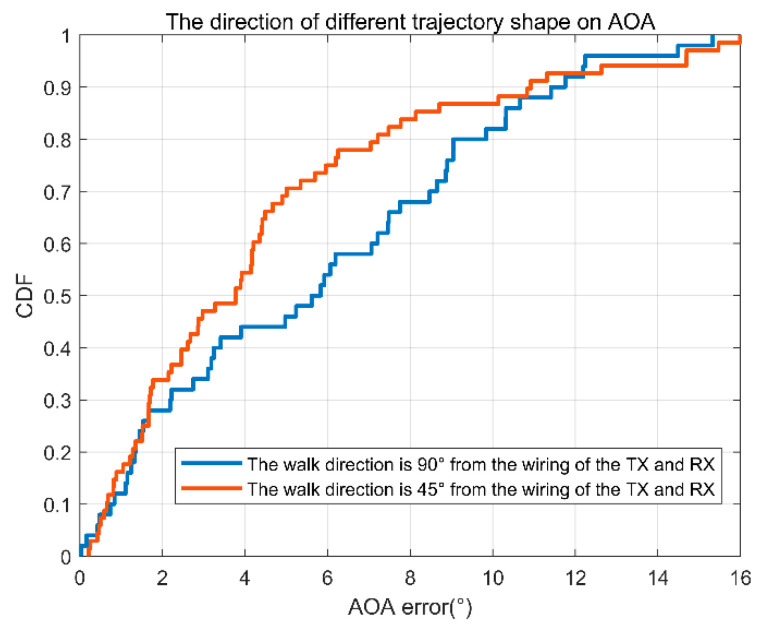
AOA error accumulation diagram of two different directions of motion.

**Figure 19 sensors-22-00276-f019:**
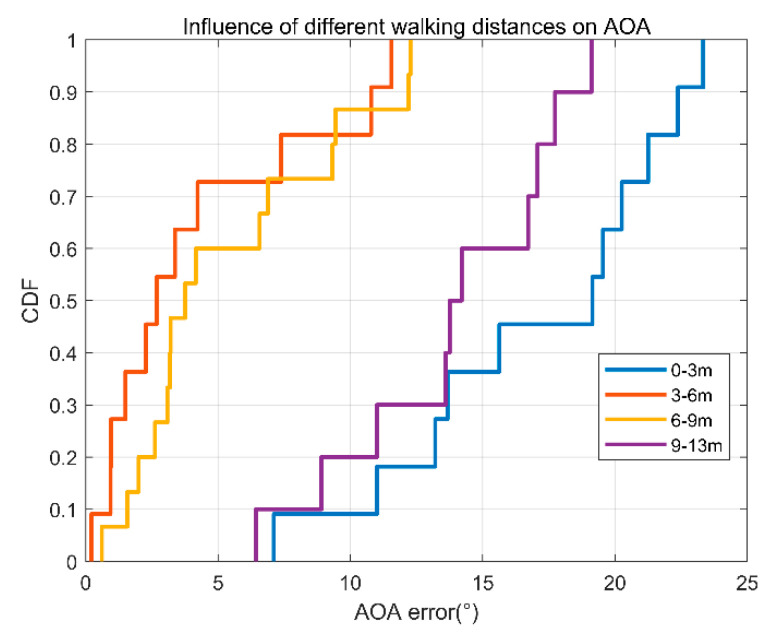
AOA error accumulation diagram of four different walking distances.

**Figure 20 sensors-22-00276-f020:**
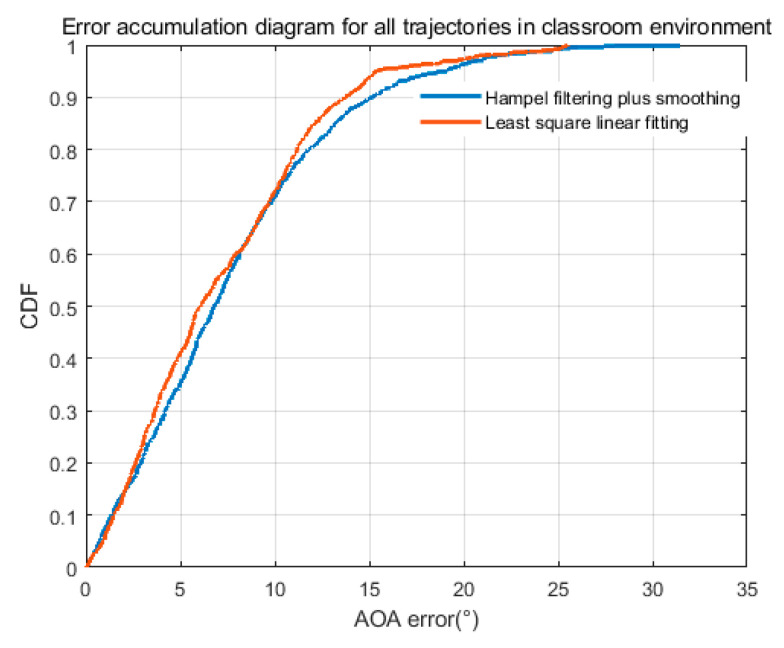
AOA error accumulation diagram of two different filtering methods.

## Data Availability

Not applicable.

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
