# Peer review of "An Angle Recognition Algorithm for Tracking Moving Targets Using WiFi Signals with Adaptive Spatiotemporal Clustering"

_sensors, 2021, doi:10.3390/s22010276_

Round 1

Reviewer 1 Report

The paper in question presents a very interesting topic: WiDar for space diversity systems in the RF band. The paper is well-written and the results are adequately supported by graphs and text-based description. My only point of revision (minor) concerns the uniqueness of each propagation path, where paths can be clustered on the basis of obstacles, whose dimensions are larger than the wavelength involved, where these clusters can be described by an appropriate shadow depth (namely, the shadow deviation), providing a log-additive expression of shadow losses (excess path loss, in general) in the RF formulae dealing with local mean power estimation/prediction [1]. These would also be complementary to localization issues in indoor environments as show in [2]. The respective presence of shadow (large-scale fading) largely influences the CDFs of the received signal for each and every antenna as show in [3].

So the aforementioned comments, references and related discussion forms an interesting aspect that the authors are kindly encouraged to include in their references and discussion, as part of a minor revision ahead of final acceptance.

[1]Salo, Jari, et al. "An additive model as a physical basis for shadow fading." IEEE Transactions on Vehicular Technology 56.1 (2007): 13-26.

[2] Palipana, Sameera, Bastien Pietropaoli, and Dirk Pesch. "Recent advances in RF-based passive device-free localisation for indoor applications." Ad Hoc Networks 64 (2017): 80-98.

[3] Chrysikos, Theofilos, and Stavros Kotsopoulos. "Characterization of large-scale fading for the 2.4 GHz channel in obstacle-dense indoor propagation topologies." 2012 IEEE Vehicular Technology Conference (VTC Fall). IEEE, 2012.

Reviewer 2 Report

The authors proposed a low level signal processing method to enhance the angle detection of non-cooperative moving objects. The proposed sub-systems sound technically reasonable, while the final results demonstrate its performance.

However, the presentation style of this paper is more like a tutorial or technical reports rather than an academical journal paper. The presentation of equation derivation is too detail and interference the main contribution. The involvement of experiment results along with the derivation also causes confusing.

the symbols and equations design should be enhanced, in the word, the current mathematic style is less convenient.

Too many errors, e.g. line 260 page 8.

I will kindly suggest authors to essentially revise the formation of their papers.

Reviewer 3 Report

The paper contains a potential valuable contribution. The subject is within the scope of the journal and the objective of research is well stated. However, some clarifications about the underlying hypothesis / scope as well as additional experiments clarifications are needed.

However, the the paper has serious flaws (e.g. figure 2 duplicated, caption duplicated)

In the opinion of this Reviewer the manuscript deserves to be published once the Author takes into account the raised issues.

Introduction / Literature review

  1. The research scope is clear as well as the literature review. Anyway, the authors should better highlight the innovative aspects of their work in the manuscript.

What are the advantages / findings in the proposed paper, which are not covered by other studies/reviews?

  1. In order to improve the literature, review this reviewer thinks that the authors should cite recently published works indoor localization (e.g. https://doi.org/10.1109/ICASSDA.2018.8477602 or https://doi.org/10.1109/TCSI.2020.2979347, paper that could be cited in the text).
  2. Please clarify the concepts of static and dynamic paths.
  3. For the sake of readability, at the end of Section 1 the authors should describe how the paper is structured.

Materials and methods

  1. Row 170: please justify the assumption that the amplitude of the static path signal is far greater than that of the dynamic path signal. As a consequence, are the authors consider an ideal scenario?
  2. Row 171: can the authors quantify the “a person moves quickly away…”?
  3. Row 180. Please recall or cite some works about the MNM signal processing algorithm to estimate AoA
  4. Do the authors consider only uniform linear spacing array? What kind of antennas are used? Please give more detail about the setup.
  5. 4. Please add the color bar values. It is impossible to understand the figure.
  6. Figure 5. Please add the legend. It is impossible to understand the figure.

Results

  1. Do the authors consider uniform linear spacing array? What kind of antennas are used? Please give more detail about the setup.
  2. What is the channel spacing? What kind of wi-fi standard are using the authors?

Minor

  1. The authors should check that all the used acronyms are explained and not repeated every time (e.g. AoA, ToF).
  2. Please use the same acronyms (e.g. not AoA or AOA)
  3. Extensive editing of English language and style required. The paper should be carefully rechecked. (e.g. blu linetooth, advnatages, Eeror, etc)
  4. Please specify the unity of measurement for each figure

Round 2

Reviewer 3 Report

Authors have properly enriched their work, by addressing each comment in a suitable way. The paper turns out to be notably improved. Please add the color bar also to fig. 3.
